# Sustained Effects on Lung Function in Community Members Following Exposure to Hazardous PM_2.5_ Levels from Wildfire Smoke

**DOI:** 10.3390/toxics8030053

**Published:** 2020-08-05

**Authors:** Ava Orr, Cristi A. L. Migliaccio, Mary Buford, Sarah Ballou, Christopher T. Migliaccio

**Affiliations:** 1Center for Environmental Health Sciences, The University of Montana, Missoula, MT 59812, USA; ava.orr@umontana.edu (A.O.); cristi.migliaccio@umt.edu (C.A.L.M.); mary.buford@umontana.edu (M.B.); sarah.dykstra@umontana.edu (S.B.); 2The Skaggs School of Pharmacy, University of Montana, Missoula, MT 59812, USA

**Keywords:** wildfire smoke, community, spirometry, health effects

## Abstract

Extreme wildfire events are becoming more common and while the immediate risks of particulate exposures to susceptible populations (i.e., elderly, asthmatics) are appreciated, the long-term health effects are not known. In 2017, the Seeley Lake (SL), MT area experienced unprecedented levels of wildfire smoke from July 31 to September 18, with a daily average of 220.9 μg/m^3^. The aim of this study was to conduct health assessments in the community and evaluate potential adverse health effects. The study resulted in the recruitment of a cohort (*n* = 95, average age: 63 years), for a rapid response screening activity following the wildland fire event, and two follow-up visits in 2018 and 2019. Analysis of spirometry data found a significant decrease in lung function (FEV_1_/FVC ratio: forced expiratory volume in first second/forced vital capacity) and a more than doubling of participants that fell below the lower limit of normal (10.2% in 2017 to 45.9% in 2018) one year following the wildfire event, and remained decreased two years (33.9%) post exposure. In addition, observed FEV_1_ was significantly lower than predicted values. These findings suggest that wildfire smoke can have long-lasting effects on human health. As wildfires continue to increase both here and globally, understanding the health implications is vital to understanding the respiratory impacts of these events as well as developing public health strategies to mitigate the effects.

## 1. Introduction

Wildfires have become a major global concern, and in the United States (US) there are hundreds of thousands to millions of acres burned [1,2]. Consequently, wildland smoke emissions are progressively being recognized as a public health concern, due to large scale wildfire fire events [3]. The increased number of these events are attributed to anthropogenic climate change, including warmer temperatures, early spring melt, and decreased winter precipitation [4]. Lightning and human ignition of excess forest fuels from years of previous fire suppression activity, as well as forest management practices, have contributed to large scale wildland fire events [5]. It has been projected that there will be an ∼50% increase in burned areas across the western US between 2009 and 2050 and future predictive models show that this area will continue to see rapidly growing fire activity with increases of 80% burned areas in the Pacific Northwest alone [6,7]. While the western states (Washington, Oregon, Montana, Idaho, California, Wyoming, Nevada, Arizona) shoulder a majority of fires/acres burned (7 million+ in 2017), the Midwest and South had hundreds of thousands of acres of wildfires in 2017. Because of fire location and prevailing wind patterns, western Montana communities in the Northern Rockies are annually inundated with smoke from increasing seasonal wildfires [8,9,10], and the region presents an opportunity to study the health effects of wildfire smoke exposures in historically at-risk communities. 

Given recent climate trends, a growing incidence of ‘historic’ fires is appearing to represent the new normal in the western US. While most fire seasons result in significant levels of exposures, the 2017 Seeley Lake region in Montana experienced an unprecedented level of smoke exposure from nearby wildfires in terms of sustained PM concentrations. Due to numerous factors, including close proximity to multiple fires and the presence of significant inversions (atmospheric conditions that trap pollution closer to the ground), community residents were exposed to EPA-designated “very unhealthy” and “hazardous” PM_2.5_ levels for 35 of the 49 days of exposure (1 August–19 September) in the summer of 2017 with a median 24 h average of 220.9 mg/m^3^ for the entire period. Historically, studies attempting to assess the health effects of wildfire smoke in local communities have focused on medical records including emergency department visits, hospital admissions, or provider visits categorized with specific ICD codes for respiratory or cardiovascular diagnoses [11,12,13,14,15]; however, the long-term human health implications of these exposures have not previously been assessed. Wildfire smoke exposure is ascribed to an average 339,000 deaths each year, and studies have reported that wildland smoke PM is associated with respiratory effects [16,17,18]. The present study addresses this gap in knowledge with two-year follow up of evaluating community members exposed to these significant levels of wildfire smoke in Seeley Lake, MT.

During the 2017 wildfires, the Missoula City-County Health Department, Division of Environmental Quality (MCCHD-DEQ) contacted the University of Montana about the exposed community. A multi-disciplinary team headed by the IPHARM (ImProving Health Among Rural Montanans) program in the School of Pharmacy at the University of Montana was assembled to enroll and screen community members in Seeley Lake, MT and for comparison, the similarly sized town of Thompson Falls, MT whose smoke exposure during the same time period was five-fold less PM_2.5_. Participants were given multiple surveys in addition to screening of health parameters: blood pressure, pulse-oximetry, and spirometry. The present study focused on respiratory effects from exposure to wildfire smoke. Analysis of full spirometry testing on community members from Seeley Lake showed significant decreases in lung function parameters up to two years post exposure, with clinically significant decreases in FEV_1_ and changes in the FEV_1_/FVC ratio indicating obstruction. The present work represents one of the first of its kind to assess and follow an exposed cohort to determine potential long-term health effects of wildfire smoke.

## 2. Methods

### 2.1. Study Design

The study was designed to enroll and assess multiple health parameters of persons living in Seeley Lake, MT area following exposure to unprecedented levels of wildfire smoke during the summer of 2017. Using the IPHARM health screening program infrastructure, five screening visits were conducted: one initial screening in 2017 within 24 h following the last day of elevated smoke, two in 2018 and two in 2019. Screening tests included spirometry, blood pressure, heart rate, oximetry, subject survey data, and collecting blood and saliva samples for later epigenetic testing. An additional cohort was enrolled and similarly screened in July of 2018 in Thompson Falls, MT, USA.

### 2.2. Study Population

The study population consisted of male and female subjects living in Seeley Lake and Thompson Falls, MT, USA during the summer of 2017. Subjects were between 23 and 85 years of age. Exclusion criteria excluded persons under the age of 18, inability to answer survey questions, or inability to perform spirometry based on the following screening questions: 1. In the last 3 months have you had a chest injury or surgery involving the eye, ear, chest, abdomen, or been hospitalized for a heart attack? 2. Do you experience hemoptysis? 3. Have you had a respiratory infection, such as flu, pneumonia, bronchitis or chest cold, in the last 3 weeks? 4. Have you ever had a pneumothorax? 5. Do you experience regular chest pain? 6. Have you ever had thoracic, abdominal, or cerebral aneurysms? An affirmative answer to any of these questions precluded a participant from the spirometry testing. Additionally, participants undergoing current treatment for hypertension were evaluated for control (i.e., <130/80) before undergoing spirometry testing. The study protocol was approved by the Institutional Review Board (IRB) at the University of Montana and participants provided informed consent. Initial study approval was obtained by the University of Montana-Missoula Institutional Review Board on 14 September 2017 (#185-17), with annual continuation approval on 23 August 2018 and 26 August 2019.

### 2.3. Particle Exposures

The MCCHD-DEQ has monitoring stations in Seeley Lake and Thompson Falls, MT. Daily PM_2.5_ levels were chronicled using the EPA NowCast method which registered PM_2.5_ concentrations for every 12 h and then calculated a weighted average of those hours. In Seeley Lake, the air quality monitor is located outside and in close proximity to the elementary school athletic field in the town and is an average of 1.755 miles from each participant’s listed address. In Thompson Falls, the monitor is located near the parking lot of the high school and is an average of 4.74 miles from the participants’ listed addresses.

### 2.4. Study Procedures

#### 2.4.1. Surveys and Participant Screening

Clinical history was obtained through surveys with special reference to smoking habits; asthma; allergies; systemic cardiovascular and respiratory diseases; ownership of wood burning stoves; COPD/Emphysema; bronchitis; and past and current health history in two weeks prior to the screening event. General physical examination included height, weight, and arterial and brachial blood pressure measurements (PulseWave analysis system; SphygmoCor AtCor Medical).

#### 2.4.2. Pulmonary Function Tests

Pulmonary function tests were performed according to NIOSH approved guidelines using a NIOSH approved spirometer, the ndd EasyOne spirometer (ndd Medical Technologies Inc., Andover, MA, USA) or the Vitalograph asma-1 monitor (Vitalograph, Inc., Lenexa, KS, USA) with testing conducted in a seated position. After screening spirometry was performed with the acceptability of each test determined by the NIOSH-certified tester and spirometry software. The EasyOne™ spirometer has an inbuilt test quality grading system (A–D, F) that provides feedback to the operator on test acceptability and repeatability. A goal was set for three acceptable tests of A or B session quality with a limit of no more than eight attempts. This corresponded to three acceptable tests with between-test repeatability of 150 mL or less, as per ATS/ERS criteria. The forced vital capacity (FVC), forced expiratory volume in one second (FEV_1_), and FEV_1_/FVC ratio was calculated automatically by the spirometer as the percentage of predicted values based on age, height, and gender as defined by NHANES III prediction equation (third National Health and Nutrition Examination Survey, 1999) [16]. In addition, the lower limit of normal (LLN) was calculated using z-scores ((measured-predicted)/standard deviation), where z-score = −1.64 (5th percentile) is defined as the LLN [17]. Spirometry quality and lung function values were compared between testing events to assess the reliability and changes in results over time.

### 2.5. Data Analysis

Intergroup differences were evaluated using three-way ANOVA followed by Sydak’s Test for comparing multiple group means while controlling type I error. Odds ratios were generated using 2 × 2 contingency tables with one factor being the presence or absence of a clinically significant decrease in FEV_1_ over a year (>30 mL) and the other factor being the presence or absence of a health attribute (e.g., asthma, allergies, etc.). The dependent variable was simply the frequency of occurrence. Fisher’s Exact test was used to test for independence between factors with a probability of Type I error set at 5 percent (two-tailed). The 95% confidence intervals for the odds ratios were calculated using the Baptista-Pike method. Significance for odds ratios was determined by the confidence interval. If the 95% CI included ‘1′ within the interval it was not statistically significant. Statistical significance of the frequency of clinically significant decreases in FEV_1_ was determined by binomial probabilities, contrasting observed frequency of clinically deceased FEV_1_ with the expected frequency for any given 1-year period.

## 3. Results

### 3.1. Cohorts

Random, volunteer participants were enrolled from two wildfire smoke-exposed communities in western Montana: Seeley Lake and Thompson Falls. Initial enrollment events were preceded by multiple methods of recruitment including flyers, community meeting announcements, online (Facebook), and word-of-mouth. In Seeley Lake, 95 participants were originally enrolled and included in the study with thirteen new participants added in 2019. Their demographics are summarized in Table 1. Sexes were fairly evenly divided between males and females (44 to 51) with almost all identifying as “white” (one participant identified as “white” and “Hispanic”). In addition, the vast majority of respondents indicated at least a minimum of a high school diploma level of education, and a distribution of household income levels with the majority in the $30,000–$75,000 range. In Thompson Falls, 24 participants were enrolled in the comparison cohort with the majority screened identifying as female (*n* = 19) and all identifying as “white”. All patients had at least a minimum of a high school diploma level of education, and a distribution of household income levels with the majority in the $30,000–$75,000 range. Demographics are summarized in Table 1.

### 3.2. Exposures

In order to describe the extent of the PM_2.5_ exposures to the affected populations, the MCCHD-DEQ data was used to record the air quality throughout Montana during the wildfire season of 2017. The MCCHD-DEQ posts the PM_2.5_ levels for the local air quality while using the EPA guidelines to determine air quality in regards to human health. Between the dates of 1 August and 19 September 2017, Seeley Lake experienced daily PM_2.5_ averages as shown in Figure 1. The daily average values between these dates was 220.9 µg/m^3^, while 35 of those days had daily PM_2.5_ averages of >150 µg/m^3^ which fell within the range of very unhealthy (150.5 to 250.4 µg/m^3^ PM_2.5_), and had a peak of 638 mg/m^3^ which exceeded hazardous levels (250.5 to 500.4 µg/m^3^ PM_2.5_). For perspective, since 2013, there were only two years (2015 and 2018) with points above the daily threshold of 35 µg/m^3^, with one of those years, 2015, showing multiple days but only a few approaching the 100 µg/m^3^ level (data not shown). In summary, the PM_2.5_ levels in Seeley Lake were very high during the 2017 season for a sustained period of time. During this same time period Thompson Falls, another community in the Northern Rockies region, located 50 miles northwest of Seeley Lake, experienced a daily PM_2.5_ average of 47 µg/m^3^, which is still above the EPA standard of 35 μg/m^3^ in the unhealthy designation (Figure 1).

### 3.3. Lung Function Assessments

#### 3.3.1. FEV_1_/FVC Decrease

A large portion of studies assessing health effects of wildfire smoke have utilized hospital medical record databases and focused on respiratory and cardiovascular ICD-10 codes as these are considered the most likely affected outcomes. To this end, full spirometry was performed and assessed as described in order to evaluate impacts of the extensive wildfire smoke PM_2.5_ exposures on lung function. At the initial visit to Seeley Lake in 2017, 59 of 95 participants were able to go through full spirometry testing. Of the 36 without full spirometry two were unable to undergo the procedure as per the screening questionnaire (recent heart attack and pneumothorax one week prior) and the rest were due to time and personnel constraints. In the following year, 2018, 38 of 42 participants who returned were able to be reassessed with full spirometry. In 2019 there were 59 of 62 participants both new and old that performed spirometry. In 2018 and 2019 participants unable to undergo spirometry were those that answered in the affirmative to any questions during screening (including pneumothorax, abdominal aneurysm, collapsed lung, recent eye surgery, and current respiratory illness). The average lung function, FEV_1_/FVC, for the total Seeley Lake cohort in 2017 immediately following the fires was 77.5% compared to the predicted average of 77.05%. However, the difference between observed and predicted changed dramatically in the subsequent years with FEV_1_/FVC in 2018 (71.6% observed; 77.35% predicted) and 2019 (73.4% observed; 76.52% predicted) (>70% is considered normal) (data not shown). Examination of the variance between sexes, the average FEV_1_/FVC values for males fell below those of females. The decrease for the population was significant with an additional significant difference between males and females (Figure 2). The data shows that in 2018 and 2019 values (FEV_1_/FVC) for both sexes fell below their predicted values. In the comparison cohort there was also a significant decrease (−5.62%, *p* < 0.001) in the observed versus the predicted values in 2018, but no statistical variance between male and female (data not shown).

#### 3.3.2. Lower Limit of Normal (LLN)

The NHANES III set of predicted equations was used for comparison purposes and the lower limit of normal was calculated using z-scores ((measured-predicted)/standard deviation). In 2017, six Seeley Lake participants fell below the LLN with one of the participants having (diagnosed) COPD. In the following year, 2018, 17 of the Seeley Lake participants had FEV_1_/FVC values that fell below normal and in 2019 there were 14 Seeley Lake participants whose values fell below normal (Figure 3). For comparison, the Thompson Falls cohort exhibited a similar percentage of participants (45.8%) with FEV_1_/FVC values below the LLN one year following the fires (Figure 3).

#### 3.3.3. Peak Expiratory Flow

Additionally, peak expiratory flow (PEF) values were compared for all three years (2017–2019) in Seeley Lake and 2018 for Thompson Falls. As shown in Figure 4, while there was a small decrease in the year following the fire (2018), there were no significant changes in PEF for the Seeley Lake or Thompson Falls cohorts over the time of the study (Figure 4). This is routinely a highly variable measure of lung function and generally multiple values are averaged over a short time period for a more accurate assessment.

#### 3.3.4. Annual FEV_1_ Decline

In Seeley Lake, from 2017 to 2018, 19 participants had FEV_1_ values that decreased more than their expected annual decline and from 2018 to 2019, 18 participants had more than their expected annual decline as shown in (Table 2). From 2017 to 2018, there were 19 participants (Table 2) that had a clinically significant decrease in FEV_1_ (>30 mL for males and >25 mL for females), which is the limit of acceptable loss of lung function per year [18]. From 2018 to 2019, there were 17 participants (Table 2) that continued to have decreased FEV_1_ values that were greater than the expected decline per year. Overall, the FEV_1_ values were decreased, with a greater effect on males, following the wildfires (Figure 5A). However, the average FEV_1_ for all participants under the age of 65 was closer to their predicted FEV_1_ averages, while the average FEV_1_ for the >65 year-old participants was significantly lower compared with the <65 values (Figure 5B), suggesting the greatest effect is on the elderly.

### 3.4. Covariates

Multiple parameters in the Seeley Lake cohort, were evaluated as potential covariates including asthma, airborne allergies, emphysema/COPD, diagnosed cardiovascular disease, or the presence of a woodstove in the home. Analyses found no associations between allergies (pollen, dust, hay fever), emphysema/COPD, or the presence of a woodstove. While more than half (55 of 89) have a woodstove in their home, of the 37 participants that returned in 2018 for a second spirometry, half of those with stoves (10 of 20) had a significant decrease in FEV_1_ (data not shown) and the average decrease was greater in the non-woodstove participants (289.3 vs. 189.7 mL). In addition, 39 of the full cohort in 2017 listed a history of allergies, but of the 37 participants that returned for a subsequent spirometry testing in either 2018 or 2019 less than half (8 of 19 and 7 of 18, respectively) had a significant decrease in FEV_1_. With only four participants indicating a history of emphysema/COPD, only one of those presented with a significant FEV_1_ decrease (data not shown). Lastly, of the participants listing a history of asthma, only 3 of 8 had a clinically significant FEV_1_ change from 2017 to 2018, but all 8 decreased from 2018 to 2019, with seven presenting with a significant change consistent with a potential long-term risk for respiratory health effects of wood smoke exposures. In the assessment of potential risks from any of these factors, none had an odds ratio indicating an increase risk. However, while not statistically significant, with an odds ratio of 0.182, asthma is approaching being a risk of a smoke-induced decrease in FEV_1_ in the long-term (i.e., after two years).

## 4. Discussion

Wildfires are a growing and significant concern globally. Most studies assessing potential health effects of exposures to the resulting smoke have focused on historical data of emergency department visits, hospital admissions, or provider visits [11,12,13,14,15]. The previous studies reported an increase in visits with ICD codes including cardiovascular and respiratory complications in the time frame following a wildfire event. In contrast, the present study was designed to evaluate and longitudinally follow a cohort of individuals in a community impacted with significant levels of smoke from wildfires. The cohort in Seeley Lake, MT is an older population (average age: 63 years) with a fairly even distribution of sexes (Table 1), while the comparison community of Thompson Falls had an average age of 59 years with the majority of patients being female. Participants were screened for inclusion, given health and demographic surveys, and underwent spirometry testing to assess potential effects on respiratory function parameters from the exposures.

Residents living in Seeley Lake, MT during the summer of 2017 were exposed to extremely high levels of PM_2.5_ (Figure 1). PM_2.5_ is a major component of air pollution and one of the criterion air pollutants designated by the EPA and has established the PM_2.5_ cutoff to be 35.4 µg/m^3^ for “unhealthy” designations. Seeley Lake had 35 consecutive days with PM_2.5_ levels at 150.5 µg/m^3^ (a designation of “very unhealthy”) and above and 9 days where levels were greater than 250.4 µg/m^3^ (“hazardous”). There were four fires burning within a 50-mile radius of Seeley Lake in 2017, contributing to the smoke exposure of the residents. The valley location of this community allowed for the smoke from the nearby wildfires to be trapped on the valley floor with temperature inversions, a weather phenomenon that occurs when cold air at night traps air pollution in a valley and prevents it from blowing away or rising higher in the atmosphere. In other recent studies examining health impacts of smoke from wildfire events, levels of PM_2.5_ had not reached the levels of those in the Seeley Lake exposure. In Australia, during a particularly significant wildfire period in 2006/2007, the daily average for the two-month timeframe was 15.81 µg/m^3^ (max. 294.95 µg/m^3^) [15]; while during the 2007 San Diego fires there was a five-day average of 89.1 µg/m^3^ (max. 803.1 µg/m^3^) [13]. Our study’s comparison community of Thompson Falls is also located in the Northern Rockies region, in the Clark Fork river valley. For the same time period in 2017 this community was also exposed to EPA designation of “unhealthy” levels of wildfire smoke (daily average of 47 µg/m^3^ PM_2.5_), however, it was 5-fold less average PM_2.5_ than Seeley Lake, MT (Figure 1). These exposures are unprecedented and have afforded researchers the opportunity to follow a cohort longitudinally.

The most likely health effects from wildfire smoke exposures are on the respiratory and cardiovascular systems. To this end studies have generally taken the form of historical evaluations of medical records and ICD codes for respiratory and cardiovascular outcomes [11,12,13,14,15]. In the 2007 San Diego wildfires, increased respiratory medical encounters were found to correlate with peak smoke periods [13]. Likewise, Alman, et al. noted an increase in hospitalizations and ED visits for cardiorespiratory codes during the 2012 Colorado wildfires [11]. These studies are able to illustrate immediate effects of these exposures, but without individual longitudinal data we cannot appreciate, or identify, long-term complications of wildfire smoke exposures. The main physiological parameter assessed in the present study was lung function via spirometry testing. Spirometry is a common method of assessing pulmonary function that can be used to diagnose asthma, chronic obstructive pulmonary disease (COPD) and other respiratory pathologies. Spirometry is often used to evaluate lung physiology instead of X-rays and CAT scans because it can detect abnormalities in lung function even when no signs or symptoms of a disease are evident. When assessing lung function and the potential effects of environmental exposures, spirometry generates multiple parameters for comparison [19,20,21]. The main values utilized from spirometry assessments include, but are not limited to, FEV_1_ (forced expiratory volume in the first second), FVC (forced vital capacity), and PEF (peak expiratory flow). The volume (FEV_1_) is compared to the FVC volume, which is the total amount air exhaled during testing, and the FEV_1_/FVC ratio is considered a reliable indicator of lung function. In analyses of these types of data, the ratios are age-matched and individuals are evaluated based on their lower limit of normal (LLN) [22,23,24]. In addition, FEV_1_ values are used to determine whether lung function is declining at a normal rate based on age. According to the Mayo Clinic, the expected annual decline in pulmonary function in FEV_1_ is 30 mL for males and 25 mL for women.

Both short-term and long-term studies of populations exposed to pollution have found significant correlation between fine particle pollutants and respiratory morbidity and mortality [25]. Exposures to PM_2.5_ have also been consistently associated with decreases in pulmonary function in epidemiological studies [26]. Participants in our Seeley and Thompson Falls cohorts were assessed by an OSHA-certified staff scientist. The data from these studies suggest a significant decrease in lung function one year following the exposure and the decrease was maintained up to two years post smoke exposure (Seeley Lake). In Seeley Lake, a decrease in FEV_1_/FVC (Figure 2) was observed in both 2018 and 2019 with a significant difference in males. In addition, there was an increase in the number of individuals that dropped below the LLN for this parameter (Figure 3). This type of change suggests an obstruction (as opposed to a restriction) that is in the category of asthma or COPD (chronic obstructive pulmonary disease). Participants were also evaluated for clinically significant annual changes in FEV_1_ (>25 mL in females and >30 mL in males) where at least half of the participants showed a clinically significant decrease in FEV_1_ with the largest average drop in the first year after exposure (Table 2). The decreased FEV_1_ values in the Seeley Lake cohort at the initial screening suggests this parameter is more sensitive (temporally) to the smoke exposure (Figure 5A), and presents in a more vulnerable population (>65 years; Figure 5B). This decrease in FEV_1_ means that the lung is restricted from filling to its normal capacity. These lung function changes, while being statistically significant are, more importantly, clinically significant as depicted in the annual decrease of FEV_1_ volumes and the increase in the number of participants dropping below the LLN for the FEV_1_/FVC ratio. In fact, the combination of these results are considered key in diagnosing obstructive changes [27]. In addition, while previous studies found that increases in 9–10 µg/m^3^ increased the risk of asthma emergency department visits following exposure [13,15], the present study suggests a long-term implication for asthmatics. Because all seven asthmatics in the Seeley Lake cohort presented with decreased FEV_1_ two years after the wildfire event, and 6 of 7 were clinically significant decreases, it suggests asthma as a risk factor for long-term complications and not just in the context of a short-term trigger.

Exposures to significant levels of wildfire smoke may result in obstructive lung pathology [28,29]. Past studies in air pollution have focused on the effects of pollutants on the airway epithelial lining and subsequent activation of the innate immune system. Recent studies have shown that ozone, a major toxic air pollutant, induces IL-33 production by airway epithelial cells that results in activation of type 2 innate lymphoid cells (ILC2) [30,31]. The ILC2 are important sources of IL-13 [32] and have been linked to asthma, an obstructive lung pathology. Additionally, studies in both firefighters and in vitro studies reported increased IL-6, a contributor to inflammatory lung pathology [33,34], in response to wildfire or wildfire smoke extract, respectively [35,36]. The firefighters were assessed in the acute phase and found increased serum levels of IL-6, IL-8 and decreased IL-10, while the lung epithelial cultures presented with increased IL-6 production in addition to other markers of COPD including dysfunction of tight junctions. Therefore, the present study showing an increase in obstructive pathology based on spirometry results suggests a model of increased acute inflammation and activation of the innate immune system from wildfire smoke that results in tissue remodeling and decreased lung function (Figure 6).

## 5. Conclusions

Wildfires are increasing globally, both in duration and frequency and the potential for long-term implications must be considered in anticipating the public health response. It is vital to understand the long-term health implications of exposure to smoke from wildfire events. The observed changes in lung function parameters in our cohort illustrate the potential for long-term adverse health effects following a significant exposure to wildfire smoke. While the event in the present study was singular in its level and duration of smoke exposure, this is not the first, nor will it be the last exposure for these communities, due to the history of wildfires in this region (Northern Rockies) [8,9,10]. While the present study has shown a significant effect on the respiratory system of individuals in wildland smoke-exposed communities in the Western United States, it is important to note that this is an older cohort that is part of an historical at-risk population. The participants in the cohorts presented with altered lung functions categorized as an obstruction (decrease in FEV_1_/FVC ratio) similar to asthma or COPD, and while not part of the design of the present study, the addition of bronchodilator testing would be key in determining the nature of the obstruction [18]. Additionally, the data suggests asthma as a potential risk factor of a longitudinal effect that warrants further research. In addition, these respiratory effects could have long-term health impacts on a variety of physiological systems. While beyond the scope of the present study, other biological systems need to be assessed for similar effects (i.e., cardiovascular, immunological) from these exposures. While the present cohort is categorized at-risk, studies have determined that mitigation strategies aimed at this group are cost-effective [37]. To expand on these present observations, future studies will need to enroll additional age groups for comparison. Developing public health strategies to mitigate the risks to communities will be paramount in protecting the well-being of the impacted populations.

## Figures and Tables

**Figure 1 toxics-08-00053-f001:**
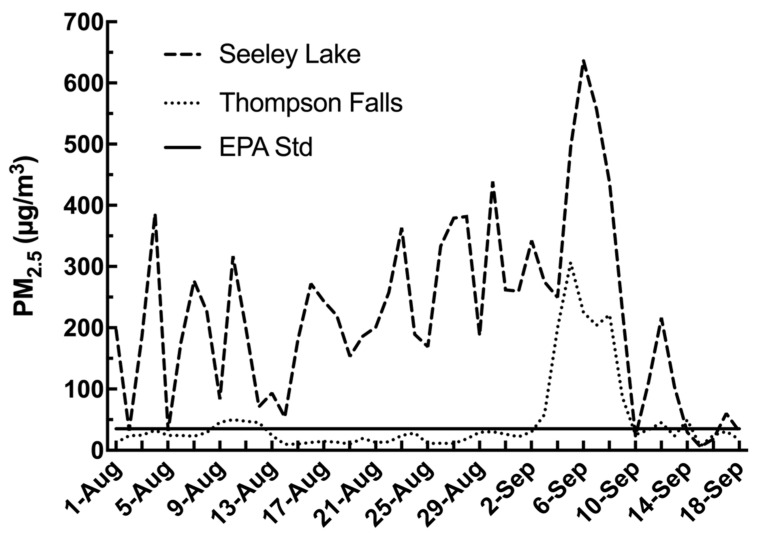
PM_2.5_ levels in two western Montana communities during 2017 fires. The above graph illustrates the PM_2.5_ levels during the peak wildfire time period for Seeley Lake and Thompson Falls. As can be seen, 2017 levels Seeley Lake were significantly above the NAAQS daily target of 35 µg/m^3^ for almost the entire period with an average of 220.9 µg/m^3^. In comparison, while Thompson Falls PM_2.5_ levels followed a similar trend and the overall daily average was 47 µg/m^3^, a large portion of the period saw daily levels below the target.

**Figure 2 toxics-08-00053-f002:**
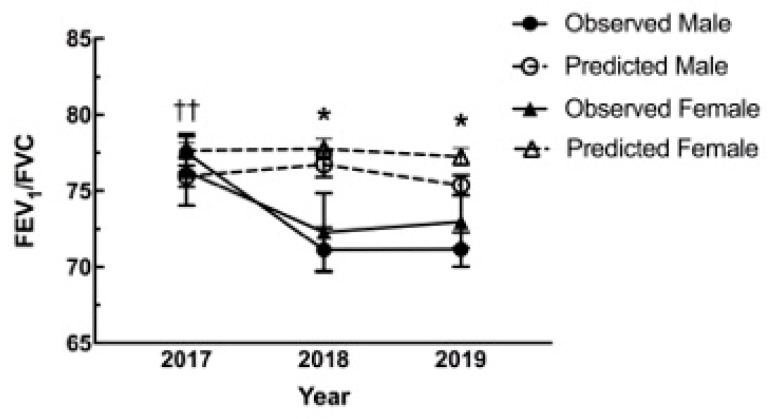
Pulmonary function changes from predicted values. The graph depicts the changes to lung function (FEV_1_/FVC) in 2018 and 2019 following the Rice Ridge fire in Seeley Lake, MT. In both years following the 2017 exposure the observed (solid lines) was significantly lower than predicted (dashed) for males (* M, circles), while significantly lower in 2019 for the females (* F, triangles). In addition, the male values were significantly lower as compared to the observed values in 2017. (* *p* < 0.05 Observed vs. Predicted within Sex; †† *p* < 0.01 Significant compared to 2017 for corresponding group).

**Figure 3 toxics-08-00053-f003:**
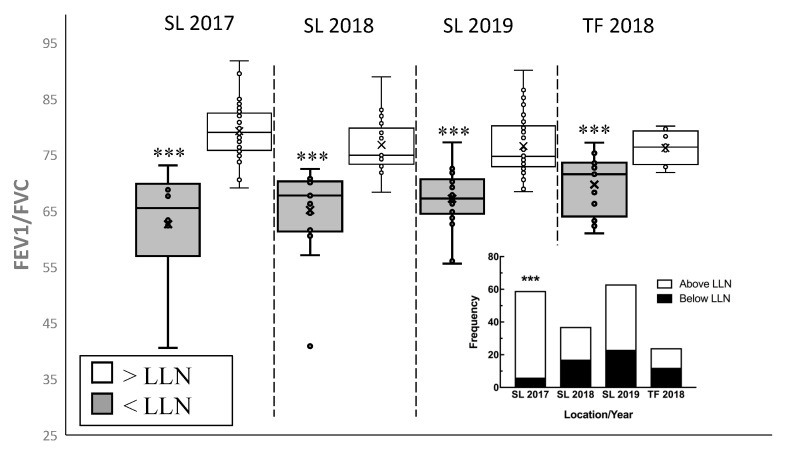
Community members falling below lower limit of normal for FEV_1_. To determine clinically significant decreases in lung function, individual FEV_1_ values are compared to the lower limit of normal (LLN) based on individual parameters (height, sex, age). LLN was calculated using z-scores (z-score < −1.64) and the following equation: (measured-predicted)/standard deviation. All three years of assessments in Seeley Lake and the one year in Thompson Falls found the average FEV_1_ values for the participants below LLN to be significantly lower that of the rest of the cohort. In addition, there is a significant increase in proportion of participants falling below the LLN, contrasted to the 2017 SL values, as shown in the inset contingency table (*n* = 24–59, *** *p* < 0.001).

**Figure 4 toxics-08-00053-f004:**
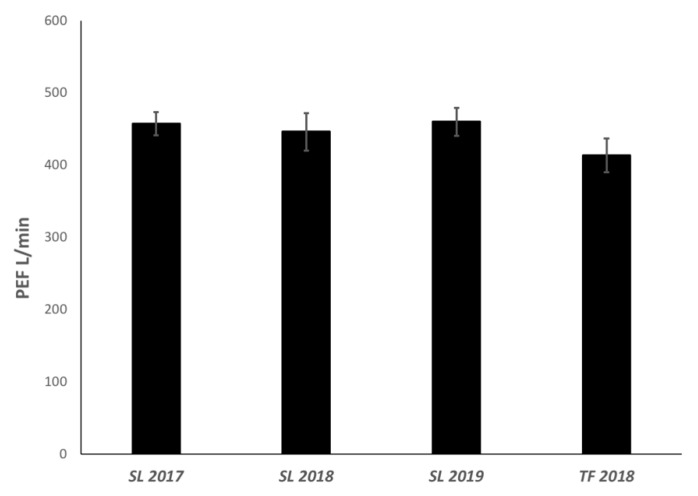
Peak expiratory flow values in the Seeley Lake and Thompson Falls cohorts. In the above graph the average peak expiratory flow (PEF) was calculated (±sem) for each year following the fires in 2017. Despite a slight decrease one year post exposure in 2018, there were no significant changes to the aggregate PEF values for the cohort and no difference with the control cohort in Thompson Falls.

**Figure 5 toxics-08-00053-f005:**
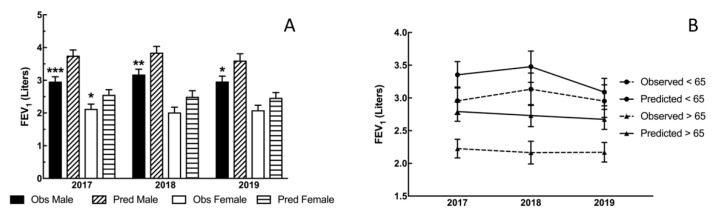
Annual changes between sex (**A**) and age (**B**) in FEV_1_ in the Seeley Lake Cohort. Changes to FEV_1_ from predicted values. The graph in panel A depicts the deviations from predicted of observed FEV_1_ values in 2017, 2018, and 2019 following the Rice Ridge fire in Seeley Lake, MT. In all three assessments following the exposure the observed FEV_1_ values were lower than predicted based on age, sex, race, and height (NHANES III–Hankinson 1999), and significantly lower all years for males and in 2017 for female participants. In addition, the male values were significantly lower as compared to the observed values in 2017. (** *p* < 0.01; *** *p* < 0.001 Observed vs. Predicted within Sex). The effect of age on FEV_1_ changes is in panel B. The observed FEV_1_ values versus the predicted FEV_1_ for the Seeley Lake, MT cohort for the three visits (over two years) following the wildfires. The two groups depicted in the graph above are categorized as either >65 years old or <65 years old. All FEV_1_ values were analyzed on individual parameters (age, sex, height) for each year. Both sets showed decreased, compared with predicted values, FEV_1_ values. The younger group (<65 years) was not significantly lower and appears to approach predicted levels by the second year after the fires; while the older (>65 years) group remained significantly lower observed values for the >65 group (* *p* < 0.05).

**Figure 6 toxics-08-00053-f006:**
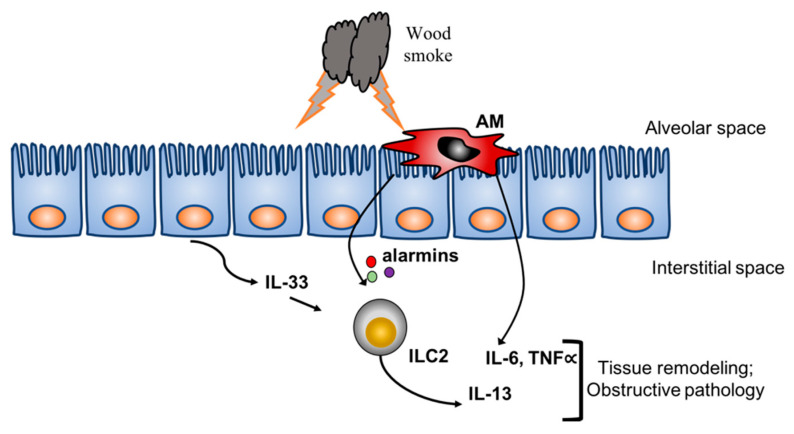
Theoretical molecular mechanism of the effects of wildfire smoke on pulmonary function. The working hypothesis is that both alveolar macrophages (AM) and lung epithelial cells directly interact with smoke particles. The combined responses result in production of key cytokines and alarmins that activate innate lymphoid cells (ILC2) and lung parenchyma resulting in tissue remodeling and an obstructive pathology (i.e., COPD, asthma).

**Table 1 toxics-08-00053-t001:** Patient demographics.

Variable	Seeley Lake			Thompson Falls
	2017	2018	2019	2018
Participants	95	42	62	24
Age * (years)	63 ± 1.5	63 ± 2.1	64 ± 1.5	59 ± 2.5
**Sex**				
Male	44	18	26	5
Female	51	24	36	19
**Race**				
White	93	40	60	24
Asian	1	1	1	0
African American	0	0	0	0
Hispanic	2	1	2	0
**Education**				
Less than High School	2	1	1	0
High School Diploma or GED	25	11	14	5
Some College	27	12	19	7
College Degree	41	16	26	12
**Income**				
Less than $29,999	18	9	11	5
$30,000–$74,999	52	22	28	15
Greater than $75,000	18	11	18	3

* Data are given as mean with ± SE; * Not all participants completed all surveys.

**Table 2 toxics-08-00053-t002:** Annual change of FEV_1_ in Seeley Lake cohort—males vs. females.

		*n*	Clinically Decreased (%)	Average Decrease (mL)
2017–2018	Total	37	19 (50%)	−0.231 ± 0.056
	Males	18	8 (44%)	−0.289 ± 0.114
	Females	19	11 (55%)	−0.208 ± 0.060
2018–2019	Total	30	17 (57%)	−0.123 ± 0.029
	Males	14	7 (50%)	−0.172 ± 0.043
	Females	16	10 (63%)	−0.135 ± 0.025

For the above table, “clinically decreased” is defined as a decrease of at least 25 mL for females and 30 mL for males over one year. The ‘*n*’ correspond to the number of participants that have full spirometry results for both of the years indicated and the first number is for the period 2017–2018 and the second for the period 2018–2019.

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
