# Peer review of "Sustained Effects on Lung Function in Community Members Following Exposure to Hazardous PM2.5 Levels from Wildfire Smoke"

_toxics, 2020, doi:10.3390/toxics8030053_

Round 1

Reviewer 1 Report

General Comments:

The overall objective of this observational study is to determine the long-term health impacts of acute wildfire smoke exposure in a human population. Acute health effects of smoke exposure are most frequently measured based upon emergency room room/physician visits soon after wildfire events, however it is unknown whether symptoms persist in a chronic fashion. This study is opportunistic in that it was initiated in response to a severe Montana wildfire event in 2017 and focuses on an older population of residents living in the area. It is not made clear whether the subjects in the study group physically remained in the area throughout the entire assessment period.

Because the study relies on an unplanned event, the analysis is limited to subjects recruited shortly after the resolution of the 2017. As a “control” group, the authors recruited subjects from a nearby community that received a relatively reduced level of exposure. Follow-up spirometry of subjects over a 3-year period showed an overall reduction in lung function which continued to decline with time. The study is limited in that the age group was exclusively a population that is considered more susceptible to health outcomes following pollutant exposure. Moreover, there is limited exposure data provided outside of the 2017 event. However, this observational study is important due to the finding that respiratory responses do not readily revert to “normal” following severe exposure events.

Major Comments:

  1. Were the human subjects living in either Seeley Lake or Thompson Falls exposed to other wildfire events prior to 2017? Is it feasible to estimate cumulative PM2.5 exposures for the subjects based upon how long they have lived in their respective communities?
  2. Was the wildfire smoke PM2.5 chemically characterized?
  3. Was the wildfire smoke primarily from biomass combustion or did burning homes (e.g. plastics, electronics) also contribute?
  4. It would be important to comment on the limitations of evaluating parameters within an older population of subjects within the discussion.

Reviewer 2 Report

The manuscript is well written and the thoughts and rationale well presented. Overall, this work insightfully describes the immediate and long-lasting effects of wildfire smoke exposure and lung function. In addition, it provides preliminary evidence linking exacerbation of lung function decline in subjects with asthma. Few comments need to be addressed before publication:

  1. What do the authors exactly mean in line 51, when they state that “…PM2.5 levels for 35 of 49 consecutive days in the summer of 2017”? That individuals were consistently exposed for period ranging from 35-49 days? Or that, while the PM2.5 levels were consistently high for 49 days, individuals were exposed for at least 35 consecutive days? A similar description is clearer in lines 164-165. In addition, the authors mention that the dates of interest Aug 1st and Sep 10th, which is a 41-day window. Are the 49 days also counting dates prior/after those described in the manuscript? Please clarify both statements.
  2. The authors describe inversion in the manuscript. Do the authors think that a description of the geography of the two areas sampled in the manuscript would be helpful? Are the communities living in a basin/valley where this particulate could reach hazardous levels for lengthy periods of time?
  3. In the study design the authors mention that two follow-up visits were performed in 2018 and 2019. Were those (or at least one) performed in the summer time as well? Weather could also affect the performance of those pulmonary function testing studies.
  4. Are the new participants (included in 2019) also residents of Seeley Lake at the time of the wildfire?
  5. Do the authors have employment information for the subjects? For instance, did some individuals have a predominantly outdoor/indoor job which would increase exposure during the sampled periods? This would help linking individual lung function results to burden of exposure.
  6. Please include the acronym LLN in line 212 when describing the Lower Limit of Normal, as this is the first time the term is introduced in the manuscript.
  7. Line 224 a parenthesis is missing
  8. Since the Thompson Falls cohort is predominantly female, have the authors looked and compared with the Seeley Lake female cohort? It would be interesting to see if there are sex-matched effects based on PM2.5 exposure (since TF [x5] > SL).
  9. Did any of the people who came back for follow-up visits relocated to other areas between 2017 and 2019? If yes (and if it is a sizable number), were the pollution levels in the new area different from the average SL concentrations?
  10. Line 341. I am unsure if 3 year qualifies to name the study long term. Perhaps mid-term is a better fit.
  11. Did any of the study subjects die between 2017 and 2019? If so, were any of the causes of death attributable to pulmonary injury described in this study?
